# Intention to Use Primary Healthcare Services among South–South Migrants

**DOI:** 10.3390/ijerph21091258

**Published:** 2024-09-23

**Authors:** Consuelo Cruz-Riveros, Alfonso Urzúa, Carolina Lagos, Evelyn Parada

**Affiliations:** 1Escuela de Enfermería, Universidad Santo Tomás, Concepción 3349001, Chile; eparada2@santotomas.cl; 2Escuela de Psicología, Universidad Católica del Norte, Antofagasta 1270709, Chile; alurzua@ucn.cl; 3Escuela de Enfermería, Universidad Santo Tomás, Santiago 8370003, Chile; clagos17@santotomas.cl

**Keywords:** transients and migrants, primary healthcare, perceived discrimination, health services

## Abstract

(1) Background: To evaluate a model based on the right-to-health approach, considering the impact of associated factors on the future utilization of primary healthcare services among international migrants in Chile. (2) Methods: A cross-sectional design was employed to survey 499 South American migrants residing in Chile. Ad-hoc questionnaires were used to assess their experiences related to the right to health, perceived discrimination, income, education, length of residence, age, marital status, gender, migration status, among others. Correlation analyses were conducted, followed by path analysis with significant variables to assess the fit of two models. (3) Results: Ten variables were identified as significant for path analysis. Among the two evaluated models, the final model identified six variables with significant direct and indirect effects. Among them, the availability, accessibility, acceptability, and quality of healthcare services were positively associated with the future intention to use them. Additionally, perceived racial and ethnic discrimination also had a positive effect on the intention to use healthcare services, suggesting a possible adaptive response to adversity, exhibiting acceptable goodness-of-fit indices (χ^2^ =241,492; *p* < 0.001; CFI = 0.913; TLI = 0.82; RMSEA = 0.062; SRMR = 0.05). (4) Conclusions: While the initial model provides valuable insights, it is essential to broaden the analysis to include other factors influencing the specific context of international migrants.

## 1. Introduction

International migration is defined as the movement of people across a country’s borders to settle temporarily or permanently in another country [1]. According to the United Nations (UN), the total number of international migrants worldwide reached 281 million in 2020 [2]. In the same year, there were 15 million migrants in Latin America and the Caribbean, with countries such as Argentina, Chile, and Brazil being preferred destinations for migration, mainly from Andean countries like Bolivia, Peru, and Ecuador, as well as Venezuela [3].

Migration is a complex phenomenon that has significant implications for the health of migrants and the host communities [4], especially when situations arise that may affect the exercise of the so-called right to health. The right to health is a fundamental principle recognized internationally, stating that every person has the right to the highest attainable standard of physical and mental health [5], which has important implications for the protection of the human rights of migrants [6]. The right to health is supported by international human rights law, which includes the Universal Declaration of Human Rights, the International Covenant on Economic, Social, and Cultural Rights, and the Convention on the Rights of the Child, among other instruments [5,7,8,9]. These instruments recognize and protect the right to health of all people, including international migrants, without any discrimination [7,8]. States have the responsibility to ensure and protect the right to health of international migrants, which implies guaranteeing access to adequate and culturally appropriate healthcare services, as well as the prevention and treatment of diseases and the protection of migrants’ mental health. Moreover, migration policies and practices must be consistent with human rights and should not hinder migrants’ access to health services [7,8].

The exercise of the right to health can be compromised for international migrants due to a series of obstacles that hinder the utilization of health services, including discrimination, language barriers, cultural barriers, lack of documentation or health insurance, and restrictive immigration policies that prevent access to these services [4,10,11,12]. Additionally, migrants may face specific health risks due to precarious living conditions during the migration process, exposure to infectious diseases, and mental health problems caused by stress and anxiety related to migration [12,13]. This health risk is increased by irregular migration, a significant phenomenon in Latin America, given the high presence of migrants facing risks and challenges associated with lack of documentation and migration outside regular channels [10]. This may include exposure to labor exploitation, human trafficking, violence, and other dangers in accessing healthcare services, as well as legal, cultural, economic, and social barriers [4].

In the context of health service utilization, two crucial factors play a significant mediating role: economic income level and education level [14,15]. A higher economic income level facilitates the use of higher quality health services with greater ease, overcoming financial barriers that often limit such access [16]. Moreover, this factor also influences the ability to access more comprehensive and timely healthcare [16].

Several additional factors have also been identified that influence the utilization of health services, such as age, sex, length of residence, type of employment, marital status, and prejudice [17,18,19,20,21,22]. Regarding age, younger migrants are more likely to use health services than older ones due to their greater healthcare needs [17]. In terms of sex, women tend to use health services more frequently than men, as they may have additional medical needs related to reproductive health and pregnancy [17]. Concerning the length of residence in the host country, individuals who have lived there longer are more likely to use health services, as they may be more familiar with the healthcare system and have established a relationship with a healthcare provider [15].

The type of work performed by migrants is also related to the need for healthcare, as certain jobs may expose individuals to risks or physical demands that increase the likelihood of requiring medical attention [15]. Marital status also influences healthcare utilization, as married individuals or those with partners and children may have greater healthcare needs related to family health, making them more likely to seek healthcare services [15,18].

Finally, international migrants may face significant barriers to healthcare utilization, including prejudice, discrimination, and administrative issues in healthcare centers [15,19]. This study focuses on South–South migration, specifically migrants from one South American country to another, in this case, Chile. There is evidence of negative factors that may affect the exercise of the right to health, such as discrimination [15,18,19,20,23], which we have included as a variable. In this context, the present study aimed to identify a model based on the right-to-health approach that allows for the evaluation of factors associated with the potential utilization of primary healthcare services for migrants in Chile. The proposed hypotheses imply an association between the future utilization of primary healthcare and variables such as economic income level, age, sex, length of residence, employment activity, educational level, marital status, migration status, exercise of the right to health (availability, accessibility, acceptability, and quality), and perceived discrimination.

## 2. Materials and Methods

### 2.1. Design

An analytical cross-sectional design was employed to assess the mediation of factors related to the potential utilization of primary healthcare services.

#### Participants

The participants were 499 international migrants residing in the regions of Antofagasta, Metropolitana, and Del Biobío (north, center, and south of Chile). The inclusion criteria were being at least 18 years old, speaking Spanish, having lived in the country for more than six months, and having used a primary healthcare service. The exclusion criteria were the presence of cognitive health problems such as dementia and other pathologies that prevent the understanding of the applied survey. The sampling type was non-probabilistic, and snowball sampling and convenience sampling were used to complete the sample. Participation was voluntary, and there was no incentive.

### 2.2. Instruments

Sociodemographic questionnaire: Variables recorded included economic income level, age, sex, length of residence in Chile, employment activity, educational level, marital status, and migration status (regularized/non-regularized).

Scale of experience in the exercise of the right to healthcare: The EERHC scale [24] was used, consisting of 25 items grouped into four dimensions: availability with 8 items, accessibility with 5 items, acceptability with 6 items, and quality and safety with 6 items. For the analyses, the mean score of each dimension will be used to represent the level of experience in the exercise of the right to healthcare. Each item was rated on a 1 to 4 scale (1 “completely disagree”, 2 “disagree”, 3 “agree”, and 4 “completely agree”). The adjustment indices indicate that the model is adequate. The reliability of the scale was estimated with Cronbach’s α coefficient = 0.98 and McDonald’s ω = 0.98.

Perceived discrimination: The scale developed by Krieger et al. [25], consisting of 9 items, was used to assess perceived discrimination among migrants. The scale has two dimensions: racial discrimination and ethnic discrimination. This scale has been used in previous studies in Chile, demonstrating optimal reliability levels (ω > 0.90) [25,26]. The instrument scores range from 1 to 4 (1 “Never”, 2 “Once”, 3 “Two or three times”, and 4 “Four or more times”). For the analyses, the mean score of each dimension will be used to represent perceived discrimination levels.

### 2.3. Future Utilization Perception

Participants were asked how likely they were to visit a primary healthcare facility in case of need. Response options were 1 “strongly disagree”, 2 “disagree”, 3 “neither agree nor disagree”, 4 “agree”, and 5 “strongly agree”.

### 2.4. Ethical Considerations

The study was approved by the Ethics Committee of Universidad Católica del Norte (resolution 015/2021). All participants signed the voluntary informed consent for the study. The study adhered to the principles of voluntariness, confidentiality, and anonymity of the participants, as reflected in the signed informed consent.

### 2.5. Data Analysis

For data analysis, SPSS version 24 was used for descriptive statistical analysis, including mean, frequency, and percentages, as well as the Kolmogorov–Smirnov normality test, T-tests, analysis of variance (ANOVA), and Spearman’s correlation test. The calculations were based on the means of the dimensions of the perceived discrimination instrument (racial and ethnic discrimination) and the means of the dimensions of the scale of experience in the exercise of the right to healthcare (availability, accessibility, acceptability, and quality). Path analysis was conducted using Mplus, chosen for its ability to estimate the magnitude and significance of hypothesized associations when single-item measures or the average of a scale are available [27]. The model fit was interpreted according to fit indices, using cut-off points such as CFI > 0.90, TLI > 0.90, RMSEA < 0.08, and SRMR < 0.08 [28,29]. This approach facilitated the evaluation of relationships between latent and observed variables, providing a deeper understanding of the factors influencing the utilization of primary healthcare services among international migrants. This estimator is particularly suitable when the data do not meet the assumptions of multivariate normality, which is common in studies involving categorical responses. Additionally, WLSMV does not assume a normal distribution of errors, making it robust against violations of this assumption. The use of WLSMV allows for more accurate and reliable estimates in contexts where the observed variables are ordinal, as is the case in our study. The descriptive analysis section was expanded to provide a detailed description of the procedures and results presented in each table.

## 3. Results

### 3.1. Participants

The mean age was 37.65 years (SD = 12.2), and the mean length of residence was 6.35 years (SD = 6.2). Regarding gender, 59.7% of the participants were women and 39.7% were men. In terms of employment, 63.1% were dependent workers or employed by others, and 16.8% were self-employed. Concerning educational level, 48.3% had completed secondary education, and 16.8% had completed technical studies. In terms of marital status, 55.3% were single, and 34.1% were married. Regarding migration status, 76.2% were documented (regularized) and 21.2% were undocumented (not regularized). In terms of economic income, 51.3% received between 300,000 to 600,000 Chilean pesos (360 to 720 USD), 27.9% received between 100,000 to 300,000 pesos (120 to 360 USD), and 12.6% received less than 100,000 pesos (120 USD) (Table 1).

Regarding country of origin, the sample consisted of 38.1% participants from Venezuela, 26.7% from Colombia, and 16.4% from Peru. In terms of health insurance, 90.4% of respondents were users of FONASA, the public health system in Chile, while 6% had no health insurance, and 2% had private health insurance (ISAPRE) (Table 1).

### 3.2. Predictor Variables

As shown in Table 1, participants generally expressed agreement with the services received in terms of availability, accessibility, acceptability, and quality in the health services evaluation survey (EERHC). Regarding the perceived discrimination instrument, participants had a mean score of 1.27, indicating that they have experienced few episodes of discrimination on average.

### 3.3. Perception of Future Utilization

As shown in Table 2, for the total sample (n = 499), the mean for the future utilization variable was 4.11 (SD = 0.91). No significant differences were observed in the means between women (M = 4.16; SD = 0.89) and men (M = 4.04; SD = 0.92), nor between individuals with regularized migration status (M = 4.11; SD = 0.92) and those without regularized status (M = 4.15; SD = 0.88). The analysis began by examining the equality of variances using Levene’s test for sex, which yielded an F value of 0.05 (*p* = 0.08). This result supports the null hypothesis of equal variances between groups. Additionally, the *t*-test yielded a value of 1.47 (*p* = 0.14), indicating no significant differences in future utilization between women and men. For the migration status variable, Levene’s test showed an F value of 0.16 (*p* = 0.69), suggesting a lack of equal variances. Similarly, the *t*-test value of −0.41 (*p* = 0.68) indicates no significant differences in future utilization between individuals with regularized and unregularized migration status.

The following data were obtained through the ANOVA statistical analysis, which allows for comparing the means of several groups to determine if there are significant differences between them. The analysis displays the estimates of means and standard deviations in the data sets stratified by educational level, perceived income, employment status, and marital status of the participants. Statistically significant differences were found based on educational level (F (3.346) = 4.241; *p* = 0.000), perceived income (F (5.381) = 6.832; *p* = 0.000), and employment status (F (5.501) = 5.501; *p* = 0.000). No significant differences were observed concerning the marital status of the participants.

It was found that the mean corresponding to the stratum comprising primary or basic education is significantly higher than the mean of those with completed technical studies (*p* = 0.042) and higher university studies (*p* = 0.000). Similarly, it was observed that the mean of those with secondary or high school education is significantly higher than that of those with completed university studies (*p* = 0.003). Additionally, those with incomplete technical studies have a significantly higher mean than those with completed university studies (*p* = 0.035).

Regarding economic income, those reporting an income of less than 100,000 pesos (approximately 120 dollars) have a significantly lower mean than those with incomes between 100,000 and 300,000 pesos (*p* = 0.006) but, in contrast, show a significantly higher mean compared to the income range between 600,000 and 1,000,000 pesos (*p* = 0.000).

Regarding the employment category, it was found that only the mean of self-employed workers is significantly higher than that of those employed by others (dependent workers) (*p* = 0.000).

### 3.4. Relationships between Predictor Variables and Future Utilization

Association analyses were conducted between the independent variables age, length of residence, availability, accessibility, acceptability, quality, racial discrimination, ethnic discrimination, and future utilization of primary healthcare (PHC). The variable age was removed from the initial model as it did not show a significant correlation with future utilization. Regarding the other variables, a significant relationship was found with future utilization. This relationship for availability, accessibility, acceptability, and quality showed a positive correlation coefficient with a moderate correlation strength. In terms of racial and ethnic discrimination, a significant correlation was found, with a negative correlation coefficient of weak strength.

### 3.5. Path Analysis

Considering the inclusion of variables that were significant for future utilization in both correlation and mean difference analyses, a model was developed incorporating the domains of the evaluation of the experience in the exercise of the right to health and perceived discrimination as predictor variables, with length of residence, educational level, income level, and employment activity as mediators. Since the simultaneous presence of four variables reached the maximum variance level, it was decided to analyze two models incorporating two mediator variables at a time.

In the first model, the mediator variables of length of residence and employment activity were included. The results of this evaluation did not indicate a mediating effect between the mentioned variables and the projected future utilization of primary healthcare services, with fit indicators not reaching acceptable levels (Table 3). As a result of this finding, the mediation hypothesis in this context was discarded.

The second model is characterized by the inclusion of variables related to income level and educational level. In this model, the fit indicators found were reasonable (Table 3).

Figure 1 illustrates the path analysis model evaluating the relationships between various predictor variables (availability, accessibility, acceptability, quality, racial and ethnic discrimination), two mediating variables (education level and income level), and the dependent variable (future utilization intention of healthcare services). The following section describes and explains all the regression coefficients obtained.

**Direct relationships:** The results of the analysis indicate that several factors related to migrants’ experiences in accessing healthcare services are significantly associated with their intention to use them in the future.

*Perception of availability:* A higher recognition of the availability of healthcare services is correlated with a greater intention to use them in the future, with an estimated coefficient of 0.093 (*p* < 0.001). This finding suggests that migrants who perceive greater ease of access to healthcare services are more inclined to consider them as a viable option in the future.

*Perceived accessibility:* The perception of accessibility has a significant positive impact on the intention for future use, with a coefficient of 0.1 (*p* < 0.001). This implies that migrants who experience fewer barriers to accessing healthcare services are more likely to use them.

*Cultural acceptability:* Healthcare services that are perceived as culturally acceptable also positively influence migrants’ intention to use them, reflected in a coefficient of 0.092 (*p* < 0.001). This result highlights the importance of adapting services to cultural needs to encourage their use.

*Perceived quality:* The quality of healthcare services, measured by the perception of their effectiveness and care, shows a coefficient of 0.091 (*p* < 0.001). Migrants who perceive high-quality services are more likely to use them in the future.

*Racial discrimination:* Interestingly, the perception of racial discrimination presents a positive direct effect on the intention to use healthcare services, with a coefficient of 0.072 (*p* < 0.001). This may reflect an adaptive behavior where migrants who perceive racial discrimination are more motivated to access services that validate their rights or provide support in contexts of adversity.

*Ethnic Discrimination:* Similarly, ethnic discrimination shows a positive coefficient of 0.075 (*p* < 0.001), which is also associated with a greater intention to use healthcare services. This pattern can be interpreted as a proactive coping strategy in response to discrimination experiences.

When mediating variables such as education level and income level are introduced into the model, these direct effects decrease, suggesting mediation. However, if the direct effects remain significant after including the mediators, it would indicate partial mediation.

**Mediated relationships by education level and income level:** When mediating variables such as education level and income level are incorporated into the analysis, several significant indirect effects are observed, suggesting the presence of partial mediation:

*Education as a mediator of acceptability:* The relationship between acceptability and intention to use is partially mediated by education level, with an indirect coefficient of 0.192. This indicates that as education level increases, migrants place more value on the cultural acceptability of services, which strengthens their intention to use them.

*Education and perception of ethnic discrimination:* The perception of ethnic discrimination increases with higher education levels (coefficient of 0.202), which may reflect greater awareness of equity issues in health. This mediated relationship implies that education plays a role in raising migrants’ awareness of discrimination, thereby affecting their intention to use healthcare services.

*Education and racial discrimination:* As education levels increase, the perception of racial discrimination decreases, with a coefficient of −0.122. This effect suggests that education can mitigate the negative effects of racial discrimination on the intention to use healthcare services, meeting the criteria for partial mediation.

*Income as a mediator of availability:* Income levels also partially mediate the relationship between perceived availability and intention to use, with a coefficient of −0.146. As income increases, the perception of service availability decreases, which reduces the intention to use them.

*Income and accessibility:* Perceived accessibility is positively mediated by income levels, with a coefficient of 0.177. Migrants with higher incomes perceive fewer accessibility barriers, which increases their intention to use healthcare services.

**Interaction between education and income (coefficient between both mediators of 0.355):** The coefficient of 0.355 describing the interaction between education and income suggests a positive relationship between these two mediators. This means that education level and income level not only act separately but can also interact. This reinforces the idea that the combination of higher education and higher income can significantly improve the perception of accessibility and, therefore, the intention to use healthcare services. However, higher income and education levels trigger a lower perception of availability and acceptability.

## 4. Discussion

The objective of this article was to identify a model based on the health rights approach to evaluate the effect of factors associated with the utilization of primary healthcare services for international migrants in Chile.

The results related to the model fit indices, such as the Tucker–Lewis Index (TLI), which in our study exceeds the threshold of 0.8, suggest that the proposed model demonstrates a reasonable and consistent fit with the observed data [28,29]. This level of fit indicates that, while the model adequately captures the relationships between latent and observed variables, there are areas that could benefit from further refinement. For instance, the fact that the TLI is just above the minimum threshold may indicate that the model, while acceptable, could be improved by incorporating additional variables or refining those already present to more accurately capture the complex interactions among factors. This interpretation suggests that future research should consider potential model enhancements, evaluating different specifications that could provide a more robust fit, particularly in similar study contexts.

The observed results in the mediating variable of educational level indicate that as educational level increases, the future utilization of primary healthcare services decreases. It would be expected that individuals with higher educational levels would have greater awareness of the need for preventive care, leading to a greater interest in the future use of services. The contribution of previous research, which has observed a positive and significant association between income level and educational attainment, is important [14,15,27]. However, some barriers have been documented that prevent international migrants from validating their educational credentials in the host country, including a lack of income and complex and centralized bureaucratic processes in certain geographic areas, making the economic income earned essential for their continued stay in the host country [30,31,32].

Another explanation, however, may be associated with the phenomenon of bypassing, which is widely documented in low- and middle-income countries, where individuals with greater educational and economic resources avoid the use of primary healthcare services, opting instead to go directly to secondary or tertiary care centers [33]. Factors contributing to this phenomenon include better access to information, the perception of higher quality, greater technology, and specialization, which in turn would lead to a quicker resolution of the health issue [34,35].

The direct effects of racial and ethnic discrimination indicate that, surprisingly, a greater perception of racial discrimination is associated with an increased intention to utilize healthcare services. A possible explanation for this counterintuitive result could be that migrants who experience discrimination are more motivated to seek healthcare as a means of obtaining support or validation to assert their rights or due to a heightened level of perceived need in the face of adversity. Previous research suggests that individuals who have suffered discrimination often experience greater mental health challenges, such as stress, which may drive them to seek healthcare services to mitigate these effects [36,37,38].

While the results offer a model that could be linked to the reality of migrants in a South American country, it is essential to consider the specific context of international migrants when analyzing the utilization of primary healthcare services. This involves recognizing factors such as their migratory status, language barriers, cultural differences, and discrimination, as these can significantly impact their access to and use of primary health services. In the first phase of the analysis, some of these variables were considered, but no significant results were obtained.

However, it is important to note that the study did not fully address cultural differences, which can be much broader than those considered by the “acceptability” dimension, including the provision of services in an understandable language and cultural appropriateness [4,10].

Although social and community support networks were not studied as a variable in this work, their inclusion could provide valuable insights for future research. Previous studies have shown that these networks can significantly enhance the utilization of healthcare services among international migrants by offering not only information and guidance but also crucial emotional support. These networks act as a facilitating bridge, enabling more effective access to healthcare, especially in environments where migrants face cultural and language barriers. Considering these networks as a variable in future studies could offer a more comprehensive understanding of the factors influencing migrants’ healthcare access [39,40,41,42].

It is relevant to mention that our results highlight the importance of adopting a multifactorial approach when analyzing the utilization of healthcare services by migrants, thereby avoiding the ‘Table 1 fallacy’. By considering multiple factors simultaneously, our study demonstrates how variables such as accessibility, acceptability, and socioeconomic status interact in ways that would not be evident in a univariable analysis. This methodological strategy also allows us to unravel the ‘black box of multiple associations’, providing a more detailed understanding of the complex relationships that influence migrants’ behavior toward healthcare utilization. This approach offers a solid foundation for developing more specific and effective interventions in public health policies.

The limitations of the study related to the analysis methodology include caution in making causal inferences, as despite achieving good fit indices, some elements are missing. The data were obtained from a single measurement, which does not provide a more appropriate view of the phenomenon of primary healthcare utilization. For future research, a longitudinal study could provide more robust data to understand the different dynamics involved between individuals and the primary healthcare level.

## 5. Conclusions

The initial model, while providing valuable information, highlights the importance of expanding the analysis to more comprehensively incorporate the specific context of international migrants, including a more detailed exploration of cultural differences, to develop more inclusive approaches tailored to the needs of this international migrant population within the primary healthcare system.

## Figures and Tables

**Figure 1 ijerph-21-01258-f001:**
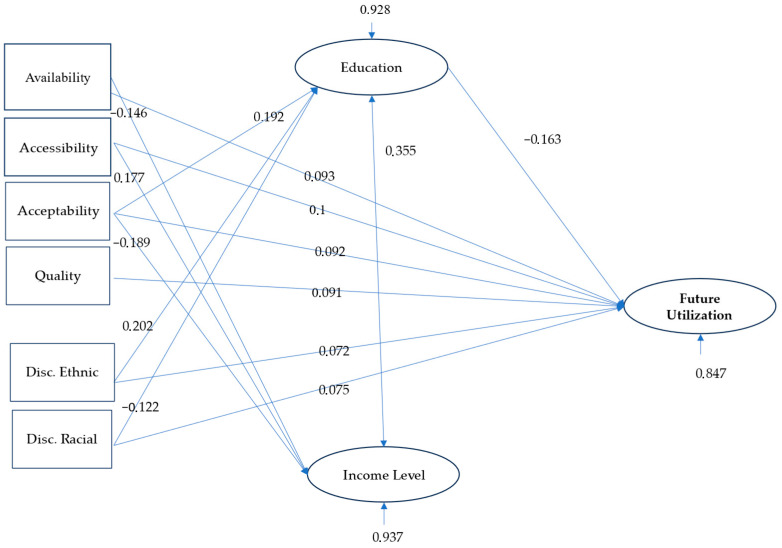
Final path model.

**Table 1 ijerph-21-01258-t001:** Descriptive statistics of variables.

	% (n)	Lower Limit	Upper Limit	Mean	Std. Deviation
Region of residence					
Antofagasta	15.6% (78)				
Metropolitana	47.1% (235)				
Del Bíobio	37.3% (186)				
Gender					
Female	59.7% (298)				
Male	39.7% (198)				
Country of origin					
Colombia	26.7% (133)				
Venezuela	38.1% (190)				
Peru	16.4% (82)				
Bolivia	10.2% (51)				
Haiti	5.2% (26)				
Argentina	1.2% (6)				
Ecuador	0.6% (3)				
Cuba	0.6% (3)				
Mexico	0.2% (1)				
Romania	0.4% (2)				
Health insurance					
Private health insurance (Isapre)	2% (10)				
Public health insurance (FONASA)	90.4% (451)				
Private	6% (30)				
Employment activity					
Self-employed or freelance worker	16.8% (84)				
Employed or contracted by others	63.1% (315)				
Retired or pensioner	2.8% (14)				
Unemployed or looking for work	6.6% (33)				
Homemaker	7.4% (37)				
Student	2% (10)				
Education level					
No education, incomplete primary education	2% (10)				
Primary education	10% (50)				
Secondary education	48.3% (241)				
Incomplete technical education	13.6% (68)				
Complete technical education	16.8% (84)				
Incomplete university education (diplomas)	1.8% (9)				
Complete university education	7.4% (37)				
Marital status					
Married	34.1% (170)				
Civil Union	5% (25)				
Single	55.3% (276)				
Widowed	2% (10)				
Separated/divorced	3% (15)				
Migration status					
Documented	76.2% (380)				
Undocumented	21.2% (106)				
Income level					
Less than 100,000 pesos	12.6% (63)				
Between 100,000 and 300,000 pesos	27.9% (139)				
Between 300,000 and 600,000 pesos	51.3% (256)				
Between 600,000 and 1,000,000 pesos	6.8% (34)				
1,000,000 pesos or more	0.6% (3)				
Scale of experience in the exercise of the right to healthcare (EERHC)					
Availability		1	4	3.11	0.62
Accessibility		1	4	3.29	0.67
Acceptability		1	4	3.21	0.62
Quality		1	4	3.23	0.61
Perceived discrimination					
Race		1	3.5	1.32	0.53
Ethnicity		1	3.8	1.25	0.47
Age		18	77	37.65	12.252
Length of residence		0	51	6.35	6.236

**Table 2 ijerph-21-01258-t002:** Correlations between independent variables and future utilization.

		Utilization Future
Age	Correlation coefficient	0.009
	Sig. (two-tailed)	0.849
Length of residence	Correlation coefficient	−0.100 *
	Sig. (two-tailed)	0.031
Availability	Correlation coefficient	0.397 **
	Sig. (two-tailed)	0.000
Accessibility	Correlation coefficient	0.319 **
	Sig. (two-tailed)	0.000
Acceptability	Correlation coefficient	0.398 **
	Sig. (two-tailed)	0.000
Quality	Correlation coefficient	0.370 **
	Sig. (two-tailed)	0.000
Ethnic discrimination	Correlation coefficient	−0.240 **
	Sig. (two-tailed)	0.000
Racial discrimination	Correlation coefficient	−0.190 **
	Sig. (two-tailed)	0.000

** Correlation is significant at the 0.01 level (two-tailed); * Correlation is significant at the 0.05 level (two-tailed).

**Table 3 ijerph-21-01258-t003:** Goodness-of-fit indices for path analysis models.

	χ^2^	Df	*p*-Valor	RMSEA IC 90%	CFI	TLI	SRMR	R^2^
Mod1	241,492	21	0.000	0.085	0.918	0.654	0.046	0.15
Mod2	241,492	21	0.000	0.062	0.913	0.82	0.05	0.15

## Data Availability

Data are contained within the article.

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
