# Peer review of "Intention to Use Primary Healthcare Services among South–South Migrants"

_ijerph, 2024, doi:10.3390/ijerph21091258_

Round 1

Reviewer 1 Report

Comments and Suggestions for Authors

The introduction covers a detailed review of existing literature and is based upon sound theoretical frame. The methods have also been well defined. Although the researchers have used non-probability sampling, it is justifiable due to a lack of listing of all migrants. It is clear that due to the approach adopted they could cover the non-documented migrants.

The bi-variate analysis (correlation in Table 3) is in line with the existing literature and easy to understand. However, table 2 does not show a consistent pattern. As the education improves, the mean of future utilisation declines. This has not been adequately discussed in the paper. Did the 'utilisation' include only 'primary healthcare facility'? There is evidence from other LMICs of a phenomenon of bypassing where the patients/ clients reach secondary or tertiary facilities. More details are needed. With increase in income, the future utilisation intention improves from first to second level but drops thereafter. This also need to be discussed.

There is a mismatch between values in text and figure 1. Some of the values in figure 1 have not been explained in the text.  The mediating role of education and income may be elaborated. Second paragraph on page 8 discusses relation between education and income but that is not the core question of the paper. How does that relationship affect future utilisation intention is not clear.

The paper discusses community and social networks but these factors had not been assessed. Based upon the study findings, do the authors have any inferences or research gap?

Author Response

Dear Reviewer, "please see the attachment" file.

Reviewer 2 Report

Comments and Suggestions for Authors

Introduction:

·         The first part of the introduction has a consistent theoretical approach. However, I consider that it is necessary to write a paragraph that emphasizes the importance of analyzing the interest phenomenon from a multifactorial approach (pathway). The “table 2 fallacy” and the “black box of multiple associations” could be mentioned.

Methods:

Design:

A non-experimental, quantitative, cross-sectional, it could be replaced by “an analytical cross-sectional”

·         Line 111  to 114: They are results.

According to the country of origin, 111 38.1% were from Venezuela, 26.7% from Colombia, and 16.4% from Peru. Regarding 112 health insurance, 90.4% were users of FONASA (public health system), 6% had no health 113 insurance, and 2% had private health insurance (ISAPRE).

2.2. Instruments

·         I recommend amplifying sociodemographic characteristics, EX: Educational level (answer 1, answer 2, answer 3).

·         The Perceived Discrimination Scale by Krieger has two dimensions, this is not specified in methods, I recommend writing it in this part.

2.5. Data analysis

·         I recommend expanding the entire descriptive analysis section. Although what was written is correct, it is a generic text that applies to any study.

·         I would recommend writing an outline of what is done in each table.

·         Although, authors described that the analysis was developed on Mplus, I recommend that to describe: It was a structural equation model based on covariance using mPlus…

·         Additionally, authors should describe the estimation method that was used  (Please check: https://www.tandfonline.com/doi/pdf/10.1080/10705511.2014.937669)

Results:

·         I feel that the results have a lot of text, and they are too many ideas. I recommend synthesizing in tables and emphasizing some specific elements.

·         Regardless of the study, I would recommend that Table 1 always be the sociodemographic characteristics of the sample. Numerous things written in table 2 could be summarized in the first table.  The first table proposed by the authors is not informative.

·         I recommend writing a table that shows for each dimension their variables and  factor loadings (magnitude and significance). This is important to verify that the measurement model has goodness of fit.

·         The final model directed acyclic graph should include all variables used. It means, each variable of different dimensions (Availability, accessibility, etc.) should be on the graph (factor loading and magnitude are optional if they are in another table).

·         It is essential that the authors review the use of elements for the graphs (Example: latent variable are in oval, observed variable are in rectangle).

·         I recommend that authors review what is the best estimation method for developing a path analysis when most variables are ordinal.

Discussion:

·         Line 248 to 250 inform the same as 95 to 97.

·         Line 252 to 255 is a goodness of fit interpretation, not discussion.

·         Line 261: I would recommend not using the term “mediator,” because direct and indirect effects were not evaluated.

·         Line 268 to 274 is a result's comment.

·         In general, among the challenges that studies with a multifactorial approach have, it is the writing of the discussion. This is because since there are few previous studies, the theory about the phenomenon from that position is short. I recommend that the authors propose possible hypotheses that explain these results and thus make the discussion. I think it is better than you comment on the results.

Round 2

Reviewer 2 Report

Comments and Suggestions for Authors

Dear authors:

I would like to respectfully recommend the following:

Line 101: is it modulation o mediation? Modulation sounds like moderation, and it was no developed.

Line 122: In the description of the Krieger scale, they do not describe the number of items that this scale evaluates, this had been recommended.

Line 139: The use of ANOVA and T-student was described in the methods, but I did not see this reflected in the analyses.

Line 146: It is not necessary to describe that it was a CB-SEM because CFI, TLI or RMSEA are indicators exclusively of CBSEM, PLS SEM uses R^2, q^2, f^2, GoF (SRMR applies for both)

Line 155-9: I think, it is not necessary.

Line 160-164, Although authors declared that they estimated indirect effects, but the are not shown. Line 273 to 285 authors describe direct effects between dimensions of Scale of Experience in the Exercise of the Right to Health Care and Perceived discrimination with Income Level. Line 287 to 294 authors describe direct effects between dimension of Scale of Experience in the Exercise of the Right to Health Care and Perceived discrimination with Income Level.

Authors could revise this reference Baron, R. M., & Kenny, D. A. (1986). The moderator-mediator variable distinction in social psychological research: Conceptual, strategic and statistical considerations. Journal of Personality and Social Psychology, 51, 1173-1182.

Line 248: Table 3 does not show CI90% for RMSEA, I recommend putting it.

Line 257 to 294: They are no indirect effects, they are just direct effects, those paragraphs say the same information on table 4 and figure 1.

Line 297: that table is named Factor Loadings by Dimension, but they are not. They are SEM Betas. When I recommended putting factor loading, I had assumed that authors have used the different dimension scales (Availability, Accessibility, etc.) as latent variable. But, re-reading I realized that they use those different dimensions as a score (maybe as a summation, and that is the reason for using correlations on table 2), so in this case, loading factors are not necessary. Perhaps, on methods, authors should clarify how each dimension was taken.  

For example, Scale of Experience in the Exercise of the Right to Health Care has 25 items, Availability was constituted by x items, they were summed.

Line 318:

As all variables are assumed as observable, they should be on rectangle. I had assumed that dimension was used as latent (constituted by each item), when you use scores come from constructs measured by items could have classification bias because it assumes that scores are totally quantitative, and they are on essence ordinal. For this, point on limitation you can comment it.
